# HIV-Related Knowledge and Sexual Behaviors among Teenagers: Implications for Public Health Interventions

**DOI:** 10.3390/children10071198

**Published:** 2023-07-11

**Authors:** Ruyu Liu, Ke Xu, Xingliang Zhang, Feng Cheng, Liangmin Gao, Junfang Xu

**Affiliations:** 1Center for Health Policy Studies, School of Public Health, Zhejiang University School of Medicine, Hangzhou 310058, China; 2Hangzhou Center for Disease Control and Prevention, Hangzhou 310021, China; 3Vanke School of Public Health, Tsinghua University, Beijing 100084, China; 4Institute for Healthy China, Tsinghua University, Beijing 100084, China; 5Institute for International and Area Studies, Tsinghua University, Beijing 100084, China; 6Department of Pharmacy, Second Affiliated Hospital, Zhejiang University School of Medicine, Hangzhou 310000, China

**Keywords:** sexual knowledge, sexual behaviors, teenager, public health interventions, implications

## Abstract

**Background:** Teenagers are at a turning point in people’s physical and psychological maturity and are also in a critical period in reproductive and sexual health. It is reported that the initial age at first sexual behavior is younger than decades ago, which implies that the risky sexual behavior among teenagers may be on the rise. However, it is unclear about the changes of sexual knowledge and behaviors in recent years. **Methods:** Based on the national sentinel surveillance survey in 2011–2021 among students in Hangzhou, we selected out teenagers aged 10–19 years as our study sample. Demographic characteristics (gender, age, marital status, etc.), knowledge of HIV and sexual behaviors were collected. The sexual knowledge score and sexual behaviors were analyzed, and their influencing factors were explored. **Results:** In total, 1355 teenagers were incorporated in this study; the awareness rates of sexual knowledge in 2011, 2013, 2014 and 2021 were 74.9%, 71.8%, 89.3% and 95.8%, respectively, which showed an overall upward trend. The results of binary logistic regression showed that the survey year, whether students had received and participated in HIV-related publicity services and whether they had sexual behaviors, had a significant influence on whether the awareness rate ≥ 75%. The survey year and whether the awareness rate ≥ 75% had a significant influence on whether students had sexual behaviors. **Conclusions:** Both the average scores and awareness rates of teenagers’ sexual knowledge showed an overall upward trend from 2011 to 2021. Teenagers’ initial sexual behavior was at a low age, and the proportion of teenagers who had fixed, temporary and commercial heterosexual sex was still relatively high despite no significant increasing. Therefore, we should further strengthen health education on the risks of sexual behaviors from schools, families and health-related institutions to ensure teenagers receive HIV-related publicity services.

## 1. Background

Teenagers are at a turning point in people’s physical and psychological maturity and are also in a critical period in reproductive and sexual health [1]. With socio-economic development and cultural and ideological changes, currently teenagers in adolescence may be becoming open to sexuality, which may increase the probability of having sexual behaviors, which may cause sexual violence, premature unwanted pregnancies and unsafe abortion [2,3,4]. For example, a history of sexual abuse during adolescence has consistently been found to be significantly associated with increased health risks and health risk behaviors in both men and women [5]. Additionally, a cross-sectional analysis about primarily students aged 13–15 years from Namibia, Swaziland, Uganda, Zambia and Zim-babwe [6] found that 23% reported having experienced sexual violence, i.e., being physically forced to have sexual intercourse at some point in their lives. These experiences were moderately-to-strongly associated with multiple adverse health behaviors, such as multiple sexual partners, poor mental health, suicidal ideation and a history of a sexually transmitted infection. For premature unwanted pregnancies, it has major effects on mother and child health, particularly in underdeveloped nations where health care is not as advanced. One of the main causes of mortality for females between the ages of 15 and 19 is pregnancy and delivery problems [7]. It also brings up a number of human rights issues. For example, a pregnant teenage girl who is forced to miss school is not only deprived of her right to education but is also prevented from her right to health at the same time because she is not allowed to access any kind of contraception or knowledge on reproductive health [7]. Additionally, now many teenagers are not physically or mentally prepared for pregnancy or delivery, which increases the risk of problems and even life-threatening health effects [8]. Moreover, risky sexual behaviors are also the main transmission route of sexually transmitted diseases, including HIV. One study has estimated that one in four sexually active teenager females has a sexually transmitted disease such as HIV, trichomoniasis or HPV [9]. Young people aged 15 to 24 are less likely to be proactive about their HIV status than older adults, which means that they may become infected with or transmit HIV [10].

The Office of AIDS Prevention Education Project for Chinese Youth released a white paper on the exploration and practice of the AIDS Prevention Education Project for Chinese Youth, which showed that nearly 3000 HIV new cases of young students aged 15–24 were reported nationwide in 2020, with sexual transmission accounting for 98.6% [11]. Similarly, UNAIDS [12] indicated that 36.7 million [32.3–41.9 million] adults aged 15 years or older and 1.7 million [1.3–2.1 million] children aged 0–14 years were living with HIV in 2021. Despite remarkable achievements in the prevention and treatment of HIV, global progress has been uneven. More than half of the world’s new infections were among women and teenagers, and nearly 2 million teenagers aged 10–19 years were living with HIV worldwide [13]. Relevant studies have also shown that somatic pubertal development [14], curiosity about sex [15], a relative lack of knowledge about sexual health [16] and accessing explicit sexual content indiscriminately on the internet [17] could increase the risk of having unsafe and unprotected sexual behaviors among teenagers.

On 29 February 2012, China’s 12th Five-Year Plan of Action for Preventing HIV was promulgated [18], requiring all general secondary schools, secondary vocational schools and general higher education schools to carry out special education activities on HIV knowledge every school year, especially for teenagers. The 13th Five-Year Plan of Action for Preventing HIV in China, dated 5 February 2017, set the work target of over 90% awareness rate of prevention and treatment among key populations including young students [19]. It can be seen that China attaches great importance to the prevention and control of sexually transmitted diseases among teenagers. However, it is unclear about the changes of sexual behaviors and HIV-related knowledge among teenagers, which could provide evidence for the future public health interventions to reduce the risk of sexual health among teenagers.

## 2. Methods

### 2.1. Participants and Data Collection

The data were collected from national HIV sentinel surveillance in Hangzhou conducted between 2011 and 2021 except the year 2012. HIV sentinel surveillance of young students was located in universities and secondary vocational colleges, and phased cluster sampling was used to collect the data during the monitoring period. Personnel specifically engaged in health behaviors monitoring are selected as the leading surveyors, and they receive strict training from the Hangzhou Center for Disease Control and Prevention, which undertakes national HIV surveillance. Everyone is also allowed to exit at any time during the filling process. Additionally, the World Health Organization identified adolescence as 10–19 years old [20], so we screened 10–19 years old from the youth sentinel and finally obtained our survey population. Among all the data, we screened out the 1355 teenagers, aged 15 years at the youngest and 18 years at the oldest. However, the psychological support for the respondents, especially the underage persons, was not assessed during the HIV sentinel surveillance investigation. The following information were collected: social demographic characteristics (gender, age, marital status, ethnicity, grade, etc.), HIV-related knowledge and sexual behaviors (drug usage, fixed heterosexual sexual behavior, temporary heterosexual sexual behavior, commercial heterosexual sexual behavior, homosexual sexual behavior, condom usage, etc.).

### 2.2. Measurement

HIV-related knowledge was measured using the 8-item HIV Knowledge Questionnaire, which has been widely applied in HIV-related surveys in China and has been proved to have good validity [21,22]. One point was given for the correct answer, and 0 points were given for the wrong answer or not knowing. Therefore, the maximum score is 8, and the minimum score is 0. Additionally, a high knowledge score indicates a good understanding of HIV transmission and prevention. The awareness rate of sexual knowledge was calculated according to the requirements of “China AIDS Prevention and Control Supervision and Evaluation Framework (Trial)” [23], which indicated that if they answer 6 or more questions correctly (≥75%), they would be considered as having knowledge awareness. In our study, sexual behaviors did not include touching but engaged in a coital relationship. Risky sexual behaviors included fixed heterosexual sex, temporary heterosexual sex, commercial heterosexual sex and homosexual sex. Fixed heterosexual sex refers to sexual behaviors with spouses or ongoing partners. Temporary heterosexual sex refers to sexual behaviors with strangers or acquaintances for once. Commercial sexual behaviors represented paying for sexual behaviors, which included commercial heterosexual behaviors and commercial homosexual behaviors. Homosexual behaviors refer to sexual behaviors with the partner who has the same gender as him or her. Unprotected sexual behaviors mean that someone did not use condom in every sexual behavior. Participation in HIV publicity services refers to the voluntary provision of relevant services as opposed to receiving publicity services. Questions asked to measure the respondents’ level of knowledge have been included in the annex.

### 2.3. Statistical Analysis

Frequency, percentage and mean ± SD were used to describe social demographic characteristics, HIV related knowledge and sexual behaviors of teenagers. Univariate analysis and binary logistic regression analysis were used to explore the factors influencing the HIV-related knowledge score and sexual behaviors. The independent variables included the survey year, gender, ethnicity, grade and whether they had received and participated in HIV-related publicity services within one year to date of the corresponding survey years. All data analyses were based on statistical software SPSS 23.0 software (IBM, Armonk, NY, USA). Variables with *p* < 0.05 were considered statistically significant.

## 3. Results

### 3.1. Basic Characteristics of Teenagers

A total of 1355 teenagers under 19 years old were incorporated in our study, with 891 (65.8%) males and 464 (34.2%) females. In terms of age, the largest number of teenagers was 17 years old (688, 50.8%), followed by 18 years old (526, 38.8%). For marital status, most (1351, 99.9%) were unmarried. Moreover, there were 511 (37.9%) teenagers in grade 1 and 797 (59.0%) in grade 2. Some 694 (52.7%) and 278 (22.9%), respectively, had received and participated in HIV-related publicity services within one year to date of the corresponding survey years (Table 1).

### 3.2. Teenagers’ HIV-Related Knowledge Scores and Awareness Rates

The average scores of teenagers’ HIV related knowledge in 2011, 2013, 2014 and 2021 were 6.25, 6.09, 6.68 and 7.22, respectively, and the awareness rates of sexual knowledge were 74.9%, 71.8%, 89.3% and 95.8%, respectively (Figure 1).

### 3.3. Sexual Behaviors of Teenagers

Among teenagers, 5 (0.4%) of them used drugs, and 83 (6.2%) had sexual behaviors before, and 1265 had (93.8%) not. The average age of their initial sexual behaviors was 16.23 ± 1.29. Among those who had experienced sex, 65 (82.3%) had their initial sexual behavior with their romantic partners, and 2 (2.5%), 9 (11.4%) and 3 (3.8%) engaged in sexual behavior with commercial sexual partners, temporary sexual partners and homosexual partners, respectively. Totals of 39 (2.9%), 45 (3.3%) and 9 (0.7%) had fixed, temporary and commercial heterosexual sex within one year to date of the corresponding survey years, respectively. For males, 1 (0.1%) had homosexual sex within one year to date of the corresponding survey years. In addition, 30 (53.6%) of the teenagers who had sexual behaviors within one year to date of the corresponding survey years had unprotected sex (Table 2).

From 2011 to 2021, percentages of those who had sexual behaviors within one year to date of the corresponding survey years were 5.25%, 5.08%, 7.14% and 2.35%, respectively. Among them, percentages of having fixed heterosexual sex were 3.4%, 3.7%, 3.6% and 1.3%, respectively, and having temporary heterosexual sex decreased from 3.6% in 2011 to 1.8% in 2021 (Figure 2).

### 3.4. Influencing Factors of HIV-Related Knowledge and Sexual Behaviors

The results of binary logistic regression showed that the survey year (*p* < 0.001, OR = 3.314), whether they had received HIV-related publicity services (*p* < 0.001, OR = 1.933), whether they had participated in HIV-related publicity services (*p* = 0.019, OR = 1.818) and whether they had sexual behaviors (*p* = 0.005, OR = 0.416) had a significant influence on the HIV-related knowledge scores. With the progress of the survey year, teenagers’ HIV knowledge scores showed an upward trend. Teenagers who had received or participated in HIV-related publicity services scored higher on HIV knowledge than those who had not (Table 3).

The results of binary logistic regression showed that the survey year (*p* = 0.004, OR = 0.174) and whether the awareness rate was ≥75% (*p* = 0.007, OR = 0.426) had a significant influence on whether they had sexual behaviors. With the progress of the survey year or the awareness rate ≥ 75%, the probability of risky sexual behaviors among teenagers would be downward (Table 4).

## 4. Discussion

In our study, the average scores and awareness rates of teenagers’ sexual knowledge both showed an overall upward trend from 2011 to 2021, like the results of previous research [24]. It may be that China attaches great importance to AIDS prevention and intervention and has achieved some success in the promoting of publicity and education on AIDS prevention and treatment [25]. Additionally, it is also possible that as the Internet grows, teenagers go online more frequently, and they will actively search the Internet for sexual-related knowledge and education, resulting in increasing accessibility of HIV knowledge to teenagers [26,27,28]. Meanwhile, in our study, the awareness rate reached 95.8% in 2021, which is the goal of the 13th Five-Year Plan of Action for Curbing and Preventing AIDS in China, that is, to reach more than 90% among key populations including teenagers. However, one study by Sun Lixiang et al. [21] on the characteristics of AIDS prevalence and results of sentinel surveillance among young students in Liaoning province found that some teenagers still had a low awareness rate of HIV knowledge, indicating that there were some misunderstandings in the population’s understanding or mastery of the characteristics of AIDS.

We consider 16.23 ± 1.29 as the average age of sexual initiation in our study to be an age at which teenagers are still considered minors in China and, according to Yan Zhang et al. [29], may expose them to increased risk of unplanned pregnancy, mental health problems or infection of sexually transmitted diseases. Additionally, there were 60 people who had sex within one year to date of the corresponding survey years, among which 30 people had unprotected sex, accounting for 53.6%, indicating that the current situation of condom use among teenagers is not optimistic, and the awareness of self-protection in the process of having sex is still insufficient, which is consistent with the findings of Liu et al. [30] Therefore, providing health education on the risk of sexual behaviors including unprotected sexual behaviors is still important. Meanwhile, carrying out intervention activities of condom promotion use [31], such as setting up condom self-service free distribution machines in schools, may help reduce the spread of sexually transmitted diseases among teenagers. Moreover, currently, teenagers’ access to prevention and treatment of sexually transmitted diseases mainly comes from some health education curriculums and less through parents and other means [32], and age-appropriate online sex education has become a more acceptable form of sex education for college students [26], indicating that there is still much room for improvement in the development of education from family members and schools. Studies have shown that teenagers without parental family members are 1.93 times more likely to engage in risky sexual behaviors than those who are living with parental family members [33], possibly because parental involvement restrains them from committing risky sexual behaviors. Thus, parents of teenagers also need to learn about relevant sexual knowledge and master parent–child communication skills to improve family sex education [34]. In addition, some studies also showed that new media could expand efficiency and coverage [35], which also allows participants to communicate and interact with each other and provides timely and convenient information feedback and thus may be more attractive to teenagers [19,36].

However, for risky sexual behaviors, the proportion of teenagers who had fixed, temporary and commercial heterosexual sex was still relatively high despite no significant increasing. Among these, temporary heterosexual sex among teenagers accounted for the highest proportion among all types of sexual behaviors. This was probably because our research population was about teenagers, not about men who had sex with men, thus the probability of homosexual sex was relatively low. Moreover, our research also found that commercial heterosexual sex also existed in the teenage population.

Teenagers who had received (*p* < 0.001, OR = 1.933) or participated in (*p* = 0.019, OR = 1.818) HIV-related publicity services within one year to date of the corresponding survey years had a higher awareness rate of sexual knowledge than those who had not, which could be that they had more exposure to HIV-related education and were more aware and familiar with AIDS. Meanwhile, risky sexual behaviors among teenagers were significantly lower in 2021 (*p* = 0.004, OR = 0.174) compared to 2011, which is different from the study of Sun Lixiang et al. [21]. This difference may be due to differences in the population of the sentinel sites included in the study. Our study included people under 19 years of age with the minimum age of 15 years (0.2%) and the maximum of 18 years (38.8%), while the Sun Lixiang’s study included people aged 15–29 years with an average age of 20.7 ± 1.6. It is also speculated that this may be due to the increase in knowledge of HIV among teenagers causing the corresponding decrease in risky sexual behaviors. Thirdly, indeed, social distancing has proven to be an active factor in controlling the spread of infectious diseases [37,38]. According to the study of Guanjian Li et al. [39], due to the COVID-19 pandemic and related containment measures, 22% of participants reported a decrease in sexual desire; 41% experienced a decrease in sexual intercourse frequency. Therefore, we have reason to believe that in the context of the outbreak of COVID-19 in 2020, maintaining social distance, avoiding mobility and reporting of geographic location carried out to prevent COVID-19 transmission [38,39,40] may also lead to lower interpersonal sexual interactions among teenagers, resulting in a decrease in the proportion of risky sexual behaviors.

## 5. Conclusions

We found that both the average scores and awareness rates of teenagers’ sexual knowledge showed an overall upward trend from 2011 to 2021; teenagers’ initial sexual behavior was at a low age, and the proportion of teenagers who had fixed, temporary and commercial heterosexual sex was still relatively high despite no significant increasing. Therefore, it is important to further strengthen health education on the risks of sexual behaviors, guide the establishment of correct relationship concepts (for example, gender relationship education should not become empty moral education or traditional “chastity” education, but rather the dissemination of scientific knowledge and social values, so that young people can gain knowledge of sexual physiology and psychology in line with their age), eliminate anxiety, ambiguity and other bad emotions in sexual development, correctly understand and deal with the morality and law of gender relations, enhance a sense of the responsibility of their own sexual behaviors, carry out intervention activities of condom promotion use, such as setting up condom self-service free distribution machines in schools, and encourage teenagers to receive HIV-related publicity services. Parents of teenagers should also be encouraged to learn about relevant sexual knowledge and master parent–child communication skills to improve family sex education. All these require the organization and publicity of policies of the health-related institutions at all levels in order to build a coordinated and synchronized whole.

## 6. Limitations

There are some limitations of this study. For example, we did not assess whether the gender of the participants was transgender of cisgender although being transgender is very rare in China. Secondly, the data collected did not cover all regions of China, and the findings may not be representative of China as a whole. In addition, by using a nationally standardized sentinel questionnaire, we were unable to explore the influences of factors not included in the questionnaire. Although the overall number of questions was the same, some questions regarding AIDS knowledge in 2021 differed from those in 2011, 2013 and 2014. Therefore, readers should be cautious when comparing the knowledge level in 2021 with other years.

## Figures and Tables

**Figure 1 children-10-01198-f001:**
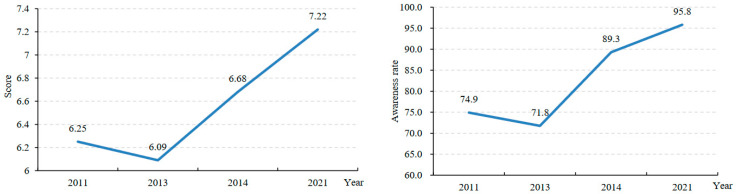
Teenagers’ HIV-related knowledge scores and awareness rates.

**Figure 2 children-10-01198-f002:**
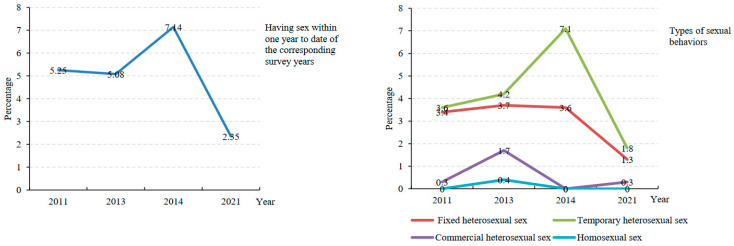
Teenagers’ risky sexual behaviors.

**Table 1 children-10-01198-t001:** Basic characteristics of teenagers.

Items		Frequency	Percentage (%)
Gender			
	Male	891	65.8
	Female	464	34.2
Age(year)			
	15	3	0.2
	16	138	10.2
	17	688	50.8
	18	526	38.8
Marital Status *			
	Unmarried	1351	99.9
	Married	1	0.1
	Cohabiting	1	0.1
Ethnicity *			
	Han	1315	97.3
	Other	37	2.7
Grade *			
	Grad 1	511	37.9
	Grade 2	797	59.0
	Grade 3 and above	42	3.1
Whether they had received HIV-related publicity services *			
	Yes	694	52.7
	No	623	47.3
Whether they had participated in HIV-related publicity services *			
	Yes	278	22.9
	No	937	77.1
Total		1355	100

Note: * indicates that 2, 3, 5, 38, and 140 people were missing for marital status, ethnicity, grade, whether they had received HIV-related publicity services and whether they had participated in HIV-related publicity services, respectively.

**Table 2 children-10-01198-t002:** Sexual behaviors of teenagers.

Items		Frequency	Percentage (%)
Drug using ^a^			
	Yes	5	0.4
	No	1295	99.6
Whether they had sexual behaviors before ^b^ (*n* = 1355)			
	Yes	83	6.2
	No	1265	93.8
Age of initial sexual behaviors (M ± SD)	16.23 ± 1.29		
The object of initial sexual behaviors ^c^ (*n* = 83)			
	Romantic partner	65	82.3
	Commercial sexual partner	2	2.5
	Temporary sexual partner	9	11.4
	Homosexual partner	3	3.8
Fixed heterosexual sex ^d^ (*n* = 1355)			
	Yes	39	2.9
	No	1305	97.1
Temporary heterosexual sex ^e^ (*n* = 1355)			
	Yes	45	3.3
	No	1299	96.7
Commercial heterosexual sex ^f^ (*n* = 1355)			
	Yes	9	0.7
	No	1335	99.3
Homosexual sex ^g^ (*n* = 891)			
	Yes	1	0.1
	No	881	99.9
Unprotected sex ^h^			
	Yes	30	53.6
	No	26	46.4

Note: ^a^, ^b^, ^c^, ^d^, ^e^, ^f^, ^g^, ^h^ indicates that 55, 7, 4, 11, 11, 11, 9, 4 were missing for drug using, whether they had sexual behaviors before, the object of initial sexual behaviors, fixed heterosexual sex, temporary heterosexual sex, commercial heterosexual sex, homosexual sex and unprotected sex, respectively.

**Table 3 children-10-01198-t003:** Influencing factors of sexual knowledge scores.

Variable		Univariate Analysis		Multivariate Analysis	
		Crude OR	*p* Value	Adjusted OR	*p* Value
Year			<0.001		<0.001
	2011 (ref)				
	2013	0.850	0.285	0.838	0.349
	2014	2.790	0.097	1.913	0.314
	2021	7.680	<0.001	3.314	<0.001
Gender					
	Male (ref)				
	Female	1.491	0.008	1.146	0.428
Ethnicity					
	Han (ref)				
	Other	1.279	0.585	1.100	0.855
Grade			<0.001		0.100
	Grade 1 (ref)				
	Grade 2	1.655	<0.001	1.065	0.741
	Grade ≥ 3	0.510	0.044	0.491	0.043
Whether they had received HIV-related publicity services					
	No (ref)				
	Yes	2.733	<0.001	1.933	<0.001
Whether they had participated in HIV-related publicity services					
	No (ref)				
	Yes	3.009	<0.001	1.818	0.019
Whether they had sexual behaviors					
	No (ref)				
	Yes	0.471	0.008	0.416	0.005

**Table 4 children-10-01198-t004:** Influencing factors of sexual behaviors.

Variable		Univariate Analysis		Multivariate Analysis	
		Crude OR	*p* Value	Adjusted OR	*p* Value
Year			0.144		0.033
	2011 (ref)				
	2013	0.966	0.910	0.628	0.206
	2014	1.387	0.665	1.473	0.627
	2021	0.434	0.030	0.174	0.004
Gender					
	Male (ref)				
	Female	0.571	0.072	0.783	0.488
Ethnicity					
	Han (ref)				
	Other	1.238	0.772	0.761	0.793
Grade			0.646		0.378
	Grade 1 (ref)				
	Grade 2	0.916	0.753	1.639	0.167
	Grade ≥ 3	1.632	0.441	1.382	0.622
Whether they had received HIV-related publicity services					
	No (ref)				
	Yes	1.032	0.907	0.979	0.953
Whether they had participated in HIV-related publicity services					
	No (ref)				
	Yes	1.948	0.024	2.909	0.005
Awareness rate ≥ 75%					
	No (ref)				
	Yes	0.471	0.008	0.426	0.007

## Data Availability

All of the main data have been included in the results. Additional materials with details may be obtained from the corresponding author.

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
