# Peer review of "HIV-Related Knowledge and Sexual Behaviors among Teenagers: Implications for Public Health Interventions"

_children, 2023, doi:10.3390/children10071198_

Round 1

Reviewer 1 Report

Thank you for this interesting paper providing some useful insights into HIV awareness of adolescents under 19 in a region of China. 

The paper is generally well written and structured, and I have only relatively minor comments. One recommendation I would make is that the abstract should be much shorter, at around 250-300 words, which is more usual for an academic paper.

Also, I note that there is no Conclusion. I recommend that a short section be added to draw your paper together and suggest any policy recommendations you may have. 

Here are my comments on each section.

BACKGROUND

This section provides a helpful background to the paper’s topic. Sources are used appropriately, and I can see that you are seeking to justify your study. It’s interesting to see how China addresses the need for HIV prevention in young people. 

There is a minor typo on line 72 – ‘incase’ should be ‘increase’. 

METHODS

This section provides a detailed overview of your methodology, tools, data sources, and analysis.  I’m not sure (line 103) what ‘fixed’ or ‘temporary’ heterosexual sex are. Can this be clarified?

On line 112, the term ‘AIDS-related publicity services’ is also a bit unclear. Are these information campaigns for the general public? Also, the phrase ‘AIDS-related’ is not usually used (AIDS is relatively uncommon due to enhanced treatment access). Would it be more appropriate to use ‘HIV-related’ and, more generally, ‘HIV awareness’, ‘HIV prevalence’, and ‘HIV prevention’?

I note that the study had ethical approval. 

RESULTS

This section provides useful details of the results. It’s interesting to see the increase in knowledge and awareness rates, and the impact suggested in the data of education about sexual health and HIV prevention. 

DISCUSSION

You draw on your findings and provide a useful summary, making some helpful points. You link this with other, similar studies. 

On line 188 you state that because the average of sexual initiation is around 16 years of age, there are additional risks. However, this is close to the global average age. If this is unusually low specifically in China, it may be worth adding a little more context here. 

On line 213 the ending of the sentence is unclear – ‘no significant increasing’. Can this be clarified? 

You make a useful point at the end of this section, about the impact of COVID-19. 

LIMITATIONS

These are useful and valid. 

REVIEWER RECOMMENDATIONS 

1.     Edit the abstract down to 250-300 words.

2.     Add a short conclusion.

3.     Address typo (line 72) and request for clarifications (line 103, line 112, line 213).

4.     Consider changing ‘AIDS-related’ (and similar) to ‘HIV-related’ (and similar).

5.     Add context in the discussion (line 188) around the age of sexual initiation being low if this relates specifically to China.

The quality of English is high, though there are some minor typos and phrasing that need correction. 

Author Response

Point 1: The paper is generally well written and structured, and I have only relatively minor comments. One recommendation I would make is that the abstract should be much shorter, at around 250-300 words, which is more usual for an academic paper.

Response 1: Thank the reviewer for the helpful suggestion. Now, we have carefully refined our abstract.

Abstract: Background: Teenagers are in a turning point in people's physical and psychological maturity, and are also in a critical period in the reproductive and sexual health. It is reported that the initial age at first sexual behavior is younger than decades ago, which implies that the risky sexual behavior among teenagers maybe on the rise. However, it is unclear about the changes of sexual knowledge and behaviors in the past years.

Methods: Based on the national sentinel surveillance survey in 2011–2021 among students in Hangzhou, we selected out teenagers aged 10–19 years as our study sample. Demographic characteristics (gender, age, marital status, etc.), knowledge of HIV and sexual behaviors were collected. The sexual knowledge score and sexual behaviors were analyzed, and their influencing factors were explored.

Results: In total, 1355 teenagers were incorporated in this study; the awareness rates of sexual knowledge in 2011, 2013, 2014 and 2021 were 74.9%, 71.8%, 89.3% and 95.8%, respectively, which showed an overall upward trend. The results of binary logistic regression showed that the survey year, whether students had received and participated in AIDS-related publicity services, and whether they had sexual behaviors had a significant influence on whether the awareness rate ≥75%; the survey year and whether the awareness rate ≥75% had a significant influence on whether students had sexual behaviors.

Conclusions: Both the average scores and awareness rates of teenagers' sexual knowledge showed an overall upward trend from 2011 to 2021; teenagers' initial sexual behavior was at a low age, and the proportion of teenagers who had fixed, temporary and commercial heterosexual sex was still relatively high despite no significant increasing. Therefore, we should further strengthen health education on the risks of sexual behaviors from schools, families, and health related institutions to ensure teenagers receive AIDS-related publicity services.

Point 2: Also, I note that there is no Conclusion. I recommend that a short section be added to draw your paper together and suggest any policy recommendations you may have.

Response 2: We thank the reviewer very much for the advice. We have now added the section of conclusion in the revised manuscript. We found that both the average scores and awareness rates of teenagers' sexual knowledge showed an overall upward trend from 2011 to 2021; teenagers' initial sexual behavior was at a low age, and the proportion of teenagers who had fixed, temporary and commercial heterosexual sex was still relatively high despite no significant increasing. Therefore, it is important to further strengthen health education on the risks of sexual behaviors, guide the establishment of correct relationship concepts, carry out intervention activities of condom promotion use, such as setting up condom self-service free distribution machines in schools, and encourage teenagers to receive HIV-related publicity services. Parents of teenagers should also be encouraged to learn about relevant sexual knowledge, master parent-child communication skills to improve family sex education. All these require the organization and publicity of policies of the health-related institutions at all levels in order to build a coordinated and synchronized whole. We have now added these in the conclusion section at the end of the manuscript.

Point 3: There is a minor typo on line 72 – ‘incase’ should be ‘increase’.

Response 3: Thank the reviewer for kindly pointing out it. We have now corrected it in the revised manuscript.

Point 4: This section provides a detailed overview of your methodology, tools, data sources, and analysis.  I’m not sure (line 103) what ‘fixed’ or ‘temporary’ heterosexual sex are. Can this be clarified?

Response 4: Fixed heterosexual sex refers to sexual behaviors with spouses or ongoing partners. Temporary heterosexual sex refers to sexual behaviors with strangers or acquaintances for once. We have now added these in the revised manuscript.

Point 5: On line 112, the term ‘AIDS-related publicity services’ is also a bit unclear. Are these information campaigns for the general public? Also, the phrase ‘AIDS-related’ is not usually used (AIDS is relatively uncommon due to enhanced treatment access). Would it be more appropriate to use ‘HIV-related’ and, more generally, ‘HIV awareness’, ‘HIV prevalence’, and ‘HIV prevention’?

Response 5: Thank the reviewer for the suggestion to further improve our study. Yes, these publicity services mainly refer not only to those conducted for the general public including students. Moreover, we have now used ‘HIV-related’ instead of ‘AIDS-related’ in the revised manuscript.

Point 6: On line 188 you state that because the average of sexual initiation is around 16 years of age, there are additional risks. However, this is close to the global average age. If this is unusually low specifically in China, it may be worth adding a little more context here.

Response 6: Thank the reviewer for the helpful suggestion. In our study, the average age at which the respondents initially had sex was around 16, which is lower than 18, when they are still minors and not yet mature enough physically and psychologically as well as face the risk of unplanned pregnancy, mental health problems or infection of sexually transmitted diseases, and still need social protection. So, we consider the average of sexual initiation is around 16 years of age as a relatively low age. We have now added these in the revised manuscript.

Point 7: On line 213 the ending of the sentence is unclear – ‘no significant increasing’. Can this be clarified?

Response 7: In our study, there was an upward trend from 2011 to 2014, but it did not show a statistical difference with the p value >0.05.

Reviewer 2 Report

Thank you for the opportunity to review the article "HIV Related Knowledge and Sexual Behaviors Among Teenagers: Implications for Public Health Interventions". It is a study that has allowed me to learn about some of the determinants related to sexual health in another context in which I usually do not work.

I have detected some inconsistencies in the article and certain aspects that are not understandable to me or that, in my opinion, require explanation:

-  - The concepts "fixed heterosexual sex, temporary heterosexual sex, commercial heterosexual behavior, and homosexual sex" are not defined. What is the difference between fixed, temporary and commercial?

        - Erratum on line 104: "ual".

-     - "Unprotected sexual behaviors represent that someone did not use condom in every sexual behavior." Why was it determined this way? In my opinion, there are non-coital/oral sexual practices in which condom use is not necessary, and should not therefore be considered "unprotected".

-       - Significance values "P<0.05" should usually appear with a lowercase "p".

-     - Basic characteristics of teenagers: I do not think it is necessary to describe in text some of the data that can be clearly seen in Table 1. On the other hand, in the age I would include the range, and the reason why this range was selected. Why was "adolescence" considered as people who fit in that age range?

-      - If there are missing values in Table 1, so that the percentages add up to 100, it is more illustrative to use the "valid" SPSS percentages.

-        - Why do the scores decline in 2013, what happened in that year?

-    - In the description of the instruments, specify the possible range of scores, i.e., between what values the scores can range. The number of items and the score are already included, but it is proper to indicate that the scores can range between X and Y.

-    - How was it assessed whether or not the adolescents had engaged in "sexual behaviors"? These behaviors can include anything from touching to a coital relationship, for example, and I think it is important to indicate this.

-    - In one of the options of "The object of initial sexual behaviors" there are 3 individuals in the category "Homosexual partner", but in the label "Homosexual sex" only one person appears. Since I can't quite understand the variables because they have not been defined, I don't know why the two values differ.

-      - In the Discussion there are statements (Internet use, mental health problems, online information, sexuality during COVID-19..., for example) that are not supported by citations, they are mere appreciations.

-      - As additional limitations, you could indicate that all participants were "apparently" cisgender (was this assessed?), mostly heterosexual, etc.

-      - Overall this is a simple study, in which it would be appropriate to justify and rely on much more citations. There are few references and a good number of them come from national institutions and projects. 

-       - Finally, I suggest reinforcing ideas about the future lines of this study.

Despite this, I would highlight the respect with which the article has worked with this population and the sociological interest it represents for the scientific community.

Reviewer 3 Report

Comprehensive review

Abstract: I would suggest shortening the abstract.

In the Introduction section:

·         The Authors have focused on the risk of HIV transmission as a consequence of risky sexual behaviour and poor knowledge, and I would recommend that they also address risks other than HIV infection (e.g. peer sexual violence, premature unwanted pregnancies, interruption of education as a result of engaging in risky sexual behaviour); the overall introduction ought to be neatly organised (may be separated into labelled parts),

In the Materials and Methods section:

·         Please describe in detail the sampling/selection of subjects from the original survey (were they also 10 year olds - this is stated in line 90 and the abstract, and no individuals of this age have been included in Table 1), the procedure for implementation of the field research, the selection of the surveyors and their qualifications, the psychological support for the respondents, especially the underage persons; was the consent of their parents given for the study of the minors?

·         I would recommend that the questions asked to the respondents (the analysed fragment of the research tool) be included in the annex, because it is unclear how certain terms were defined in the survey, e.g. sexual behaviour (whether petting, or vaginal, oral intercourse), participation in HIV prevention activities (whether e.g. receiving a leaflet or participating in a 10-hour workshop), what questions were asked to measure the respondents' level of knowledge - without this, I am not entirely convinced that the presented conclusions are valid,

In the Results section:

·         it would be worthwhile to broaden the analyses and try to highlight further tendencies rather than repeat most of the values that are contained in the tables,

·         I would advise omitting the numerical values from the analysis, as they can be found in the tables,

In Discussion section:

·         a logical discussion of the results; I would recommend that reference be made to other global studies, particularly regarding the frequency of sexual contact during the pandemic of COVID-19,

The limitations of research:

·         I appreciate the thoroughness of these limitations.

Author Response

Point 1: Abstract: I would suggest shortening the abstract.

Response 1: Thank the reviewer for the helpful suggestion. We have now shortened the abstract in the revised manuscript.

Abstract: Background: Teenagers are in a turning point in people's physical and psychological maturity, and are also in a critical period in the reproductive and sexual health. It is reported that the initial age at first sexual behavior is younger than decades ago, which implies that the risky sexual behavior among teenagers maybe on the rise. However, it is unclear about the changes of sexual knowledge and behaviors in the past years.

Methods: Based on the national sentinel surveillance survey in 2011–2021 among students in Hangzhou, we selected out teenagers aged 10–19 years as our study sample. Demographic characteristics (gender, age, marital status, etc.), knowledge of HIV and sexual behaviors were collected. The sexual knowledge score and sexual behaviors were analyzed, and their influencing factors were explored.

Results: In total, 1355 teenagers were incorporated in this study; the awareness rates of sexual knowledge in 2011, 2013, 2014 and 2021 were 74.9%, 71.8%, 89.3% and 95.8%, respectively, which showed an overall upward trend. The results of binary logistic regression showed that the survey year, whether students had received and participated in HIV-related publicity services, and whether they had sexual behaviors had a significant influence on whether the awareness rate ≥75%; the survey year and whether the awareness rate ≥75% had a significant influence on whether students had sexual behaviors.

Conclusions: Both the average scores and awareness rates of teenagers' sexual knowledge showed an overall upward trend from 2011 to 2021; teenagers' initial sexual behavior was at a low age, and the proportion of teenagers who had fixed, temporary and commercial heterosexual sex was still relatively high despite no significant increasing. Therefore, we should further strengthen health education on the risks of sexual behaviors from schools, families, and health related institutions to ensure teenagers receive HIV-related publicity services.

Point 2: In the Introduction section:

  • The Authors have focused on the risk of HIV transmission as a consequence of risky sexual behaviour and poor knowledge, and I would recommend that they also address risks other than HIV infection (e.g. peer sexual violence, premature unwanted pregnancies, interruption of education as a result of engaging in risky sexual behaviour); the overall introduction ought to be neatly organised (may be separated into labelled parts),

Response 2: Thank the reviewer for the helpful suggestion. We have now added this information in the revised section of introduction as follows. “For example, a history of sexual abuse during adolescence has consistently been found to be significantly associated with increased health risks and health risk behaviors in both men and women. And a cross-sectional analysis about primarily students aged 13–15 years from Namibia, Swaziland, Uganda, Zambia and Zim-babwe found that 23% reported having experienced sexual violence, i.e. being physically forced to have sexual intercourse at some point in their lives. These experiences were moderately-to-strongly associated with multiple adverse health behaviours, such as multiple sexual partners, poor mental health, suicidal ideation and a history of a sexually transmitted infection. For premature unwanted pregnancies, it has major effects on mother and child health, particularly in underdeveloped nations where health care is not as advanced. One of the main causes of mortality for females between the ages of 15 and 19 is pregnancy and delivery problems. It also brings up a number of human rights issues. For example, a pregnant teenage girl who is forced to miss school is not only deprived of her right to education, but is also prevented from her right to health at the same time because she is not allowed to access any kind of contraception or knowledge on reproductive health. And now, many teenagers are not physically or mentally prepared for pregnancy or delivery, which increases the risk of problems and even life-threatening health effects.”

Point 3: Please describe in detail the sampling/selection of subjects from the original survey (were they also 10 year olds - this is stated in line 90 and the abstract, and no individuals of this age have been included in Table 1), the procedure for implementation of the field research, the selection of the surveyors and their qualifications, the psychological support for the respondents, especially the underage persons; was the consent of their parents given for the study of the minors?

Response 3: Thank the reviewer for the helpful suggestion. The data were collected from national HIV sentinel surveillance in Hangzhou conducted between 2011 and 2021 except the year 2012. HIV sentinel surveillance of young students were located in universities and secondary vocational colleges, and phased cluster sampling were used to collect the data during the monitoring period. Personnel specifically engaged in health behaviors monitoring are selected as the leading surveyors, and they receive strict training from the Hangzhou Center for Disease Control and Prevention, which undertakes national HIV surveillance. Everyone is also allowed to exit at any time during the filling process. And the World Health Organization identified adolescence as 10-19 years old [20], so we screened 10-19 years old from the youth sentinel and finally got our survey population. Among all the data, we screened out the 1355 teenagers, aged 15 years at the youngest and 18 years at the oldest. However, the psychological support for the respondents, especially the underage persons were not assessed during the HIV sentinel surveillance investigation. We have now added these in the revised manuscript.

Point 4: I would recommend that the questions asked to the respondents (the analysed fragment of the research tool) be included in the annex, because it is unclear how certain terms were defined in the survey, e.g. sexual behaviour (whether petting, or vaginal, oral intercourse), participation in HIV prevention activities (whether e.g. receiving a leaflet or participating in a 10-hour workshop), what questions were asked to measure the respondents' level of knowledge - without this, I am not entirely convinced that the presented conclusions are valid,

Response 4: Thank the reviewer for the helpful suggestion. In our study, sexual behaviors did not include touching but engaged in a coital relationship. Fixed heterosexual sex refers to sexual behaviors with spouses or ongoing partners. Temporary heterosexual sex refers to sexual behaviors with strangers or acquaintances for once. Commercial sexual behaviors represented paying for sexual behaviors, which included commercial heterosexual behaviors and commercial homosexual behaviors. Homosexual behaviors refer to sexual behaviors with the partner who has the same gender with him or her. Participation in HIV publicity services refers to the voluntary provision of relevant services as opposed to receiving publicity services. We have now added these in the revised manuscript. Questions asked to measure the respondents' level of knowledge have been included in the annex.

Point 5: it would be worthwhile to broaden the analyses and try to highlight further tendencies rather than repeat most of the values that are contained in the tables, I would advise omitting the numerical values from the analysis, as they can be found in the tables,

Response 5: Thank the reviewer for the helpful suggestion. We have now refined the description of the numerical values from the analysis and highlight the potential tendencies and corresponding measures in the revised section of discussion.

Point 6: a logical discussion of the results; I would recommend that reference be made to other global studies, particularly regarding the frequency of sexual contact during the pandemic of COVID-19,

Response 6: We thank the reviewer very much for the advice. Indeed, social distancing has proven as active factor in controlling the spread of infectious diseases [37-38]. According to the study of Guanjian Li et al [39], due to the COVID-19 pandemic and related containment measures, 22% of participants reported a decrease in sexual desire; 41% experienced a decrease in the sexual intercourse frequency. Therefore, we have reason to believe that in the context of the outbreak of COVID-19 in 2020, maintaining social distance, avoiding mobility, and reporting geographic location carried out to prevent COVID-19 transmission [38-40] may also lead to lower interpersonal sexual interactions among teenagers, resulting in a decrease in the proportion of risky sexual behaviors. We have now added these in the revised manuscript.

Round 2

Reviewer 1 Report

Thank you for this revised version of the paper. I can see that you have addressed my recommendations, and I am satisfied with your edits (apart from one - see below). I have a couple of minor comments on the revised version:

1. in the discussion about the age of sexual initiation (lines 261-267) I can see that you've edited the material. It still needs a bit of tidying up - actual age does not always correlate with physical maturity and/or mental capacity to make choices. I think you can simply say that 16.23 as the average age of initiation in your study is an age at which teenagers are still considered minors in China, and, according to [citation], may expose them to increased risk of xxxx. 

2. Please confirm that your citation information is accurate - for example, I couldn't find current citation [21] in an online search. 

3. In the conclusion, on line 324, you say 'establishment of correct relationship concepts'. Can you clarify? It sounds like a value judgement as it is written. 

The quality of English language is high, with only minor discrepancies. It would benefit from a final copy edit.  

Author Response

Point 1: in the discussion about the age of sexual initiation (lines 261-267) I can see that you've edited the material. It still needs a bit of tidying up - actual age does not always correlate with physical maturity and/or mental capacity to make choices. I think you can simply say that 16.23 as the average age of initiation in your study is an age at which teenagers are still considered minors in China, and, according to [citation], may expose them to increased risk of xxxx.

Response 1: We thank the reviewer very much for the advice. We have now tidied up it. We consider 16.23 ± 1.29 as the average age of sexual initiation in our study is an age at which teenagers are still considered minors in China, and, according to Yan Zhang et al [29], may expose them to increased risk of unplanned pregnancy, mental health problems or infection of sexually transmitted diseases.

Point 2: Please confirm that your citation information is accurate - for example, I couldn't find current citation [21] in an online search.

Response 2: Thank the reviewer for kindly pointing out it. This is a Chinese citation information, and the original is “孙笠翔,周丹,赵砚,潘珊,王莉.2016-2019年辽宁省青年学生艾滋病流行特征及哨点监测结果.热带医学杂志,2022,22(08):1149-1152.”. We have now indicated it is in Chinese in the revised manuscript.

Point 3: In the conclusion, on line 324, you say 'establishment of correct relationship concepts'. Can you clarify? It sounds like a value judgement as it is written.

Response 3: Thank the reviewer for the helpful suggestion. For example, gender relationship education should not become empty moral education or traditional "chastity" education, but rather the dissemination of scientific knowledge and social values, so that young people can gain knowledge of sexual physiology and psychology in line with their age, eliminate anxiety, ambiguity and other bad emotions in sexual development, correctly understand and deal with the morality and law of gender relations, and enhance a sense of the responsibility of their own sexual behaviors. We have now added the clarification in the revised manuscript.

Reviewer 2 Report

Thank you very much for the consideration of my contributions and I wish you a lot of success with this article.

Author Response

Thank you very much for your wishes.

Reviewer 3 Report

Dear Authors, Thank you very much for the precise clarification of my concerns and the improvements which you have made in the manuscript. Your corrections have definitely made the article appear far more precise and will be of great value to a larger audience. Therefore, I congratulate you.

Author Response

Thank you very much.